# Large-scale electron microscopy database for human type 1 diabetes

Pascal de Boer [1,4], Nicole M. Pirozzi[1,4], Anouk H. G. Wolters[1], Jeroen Kuipers[1], Irina Kusmartseva[2], Mark A. Atkinson[2,3], Martha Campbell-Thompson[2] & Ben N. G. Giepmans [1✉]

Autoimmune β-cell destruction leads to type 1 diabetes, but the pathophysiological mechanisms remain unclear. To help address this void, we created an open-access online repository, unprecedented in its size, composed of large-scale electron microscopy images ('nanotomy') of human pancreas tissue obtained from the Network for Pancreatic Organ donors with Diabetes (nPOD; www.nanotomy.org). Nanotomy allows analyses of complete donor islets with up to macromolecular resolution. Anomalies we found in type 1 diabetes included (i) an increase of 'intermediate cells' containing granules resembling those of exocrine zymogen and endocrine hormone secreting cells; and (ii) elevated presence of innate immune cells. These are our first results of mining the database and support recent findings that suggest that type 1 diabetes includes abnormalities in the exocrine pancreas that may induce endocrine cellular stress as a trigger for autoimmunity.

[1] Department of Biomedical Sciences of Cells and Systems,  University of Groningen, University Medical Center Groningen, Groningen, The Netherlands. [2] Department of Pathology, Immunology and Laboratory Medicine, University of Florida Diabetes Institute, Gainesville, FL, USA. [3] Department of Pediatrics, College of Medicine, University of Florida, Gainesville, FL, USA. [4]These authors contributed equally: Pascal de Boer, Nicole M. Pirozzi. ✉email: b.n.g. giepmans@umcg.nl

The underlying mechanisms that initiate the autoimmune destruction of β-cells resulting in type 1 diabetes are still poorly understood[1], contributing to hampered efforts to prevent and/or cure the disease. Apart from pancreas or islet transplantation, exogenous insulin administration remains the only treatment for type 1 diabetes[2,3]. To improve understanding of type 1 diabetes pathogenesis and its natural history, the Network of Pancreatic Organ donors with Diabetes (nPOD) program was formed in 2007[4]. nPOD provides transplantation-quality pancreas samples recovered from organ donors including controls, patients with type 1 diabetes, and donors with type 1 diabetes-associated autoantibodies but without diabetes[5]. Most individuals with two or more islet autoantibodies will progress to symptomatic type 1 diabetes and thus the nPOD program provides unprecedented access to exceedingly rare pancreas samples[6,7]. nPOD collaboration include sharing of raw data that enables reuse and analysis by scientists worldwide analyzing the donor material to obtain new information from complex datasets in search for the trigger of type 1 diabetes[4,8].

Here, we present an open-access nanoscale image data repository of nPOD organ donor islets and report ultrastructural abnormalities and innate immune cell populations in islets and exocrine pancreas. Through large-scale electron microscopy (EM), termed 'nanotomy' for nano-anatomy[9,10], an extensive database of unbiased islets was established for 47 donors. Nanotomy enables analysis of both islets and surrounding exocrine tissue, from low to high resolution, ranging from large structures, such as the complete islet, up to macromolecules[9]. In addition, an analytical 'ColorEM' approach[11,12] further defined cell types and subcellular features based on elemental content in a label-free manner.

## Results

**Human islet nanotomy.** An open-access EM database was created, sizing in the range of >1 million traditional EM images to permit ultrastructural evaluation of human islets. A disadvantage of EM is the inherent laborious sample preparation given that islets represent only 1–2% of the pancreas volume, static non-dynamic imaging, and limited fields of view, which lack tissue context when acquiring data at high resolution. Nanotomy datasets overcome the latter problem as zoom and panning allows detailed analysis while maintaining full context of islets and acinar exocrine regions[9]. We have developed standardized nanotomy protocols (Fig. 1a), and this workflow from sample preparation of relatively large samples up to sharing via the nanotomy website, has become routine in our EM center (Fig. 1a). The nPOD nanotomy database currently contains 64 datasets from in total 47 donors including donors type 1 and type 2 diabetes, autoantibody-positive donors without diabetes symptoms, as well as from control donors. The sample quality was deemed very good and was independent of sample storage duration of up to several years. Only 1 out of 48 donor samples did not pass quality control checks for morphology. Images of complete cross sections of islets of Langerhans at macromolecular scale allow for morphological analysis of complete islets, cells, organelles, and macromolecules by simply zooming in at higher resolution within any region or feature of interest in a 'Google-earth'- like manner (Fig. 1b).

**Label-free identification of multiple parameters in type 1 diabetes donors.** Secretory cell types are readily identified based on the morphology and gray levels of the secretory granules (Fig. 1c)[13,14]. Islets are distinguishable by clustering of cells with lighter cytoplasm, smaller secretory granules, and less abundant endoplasmic reticulum (ER) than the acinar cells of the surrounding exocrine tissue[15]. Islets from control, autoantibody-positive, type 2 diabetes, and 11 of 16 type 1 diabetes donors contain the four islet cell types (α-, β-, δ-, and pancreatic poly-peptide (PP) cells). As expected from donors with type 1 diabetes, islets are often smaller and frequently lacked β-cells, absent, or only present as scattered endocrine cells through exocrine tissue.

Although identification of pancreas cell types was feasible using established granule morphological characteristics, expert assessment was still required. Further objective determination of granule content was provided using energy dispersive X-ray analysis (EDX; 'ColorEM')[11,12]. Element maps for phosphorus, nitrogen, and sulfur show the variation of granule content within each of the aforementioned cell types (Fig. 1c, see Supplementary Fig. 1 for raw element maps). Nuclei contain high amounts of nitrogen (red) and phosphorus (green) in the condensed heterochromatin giving the appearance of yellow due to red and green overlay. All secretory granules contain high nitrogen from concentration of hormones or digestive enzymes, both consisting of amino acids. Additionally, granules of human β- and mast cells contain a prominent sulfur signal because of high cysteine content and sulfur-rich proteases, respectively[13,16], while granules of α-cells are enriched with phosphorus as previously found rat α-cells[11]. Thus, analytical 'ColorEM' enables unbiased fingerprinting of granules and cell types in the vast content of gray-scaled nanotomy data.

**Heterogeneity in innate immune cell prevalence between donor groups.** Type 1 diabetes is considered a T cell driven disease; however, in these donor islets, innate immune cell infiltrates were observed including neutrophils, eosinophils, and mast cells (Table 1). Large-scale EM from nPOD nanotomy allowed quantification of each cell type per donor dataset. Eosinophils were identified intra-islet in one donor with type 1 diabetes (6243) and in the exocrine pancreas of another donor with type 1 diabetes (6380; Fig. 2a). The latter observation was also in agreement with histopathological data (Supplementary Table 2)[8]. Neutrophils were found in 9 of 16 donors with type 1 diabetes with the highest numbers in two donors with type 1 diabetes who had acute pancreatitis as assessed from histopathology (6064 and 6204; Supplementary Table 2). Except for peri-islet localization in two donors (6064 and 6436) neutrophils were found primarily in the exocrine tissue and not within islets (Table 1, Fig. 2b). Few neutrophils could be observed in the control and autoantibody-positive (3 donors) donor groups. Our finding substantiates a recent report where neutrophils and neutrophil extracellular traps (NETs) were found more abundantly in type 1 diabetes as well as autoantibody-positive donors compared to non-diabetic controls[17]. Moreover, neutrophils were typically found in the exocrine pancreas in previous immunohistochemistry[18,19] and EM studies[19]. This supports the notion that not only islets, but also the exocrine pancreas tissues are affected during type 1 diabetes[20,21].

Although mast cells were observed in every donor group, the average number of mast cells was highest, but not statistically significant, in autoantibody-positive and type 1 diabetes donors compared to control (Fig. 2g). Moreover, stronger differences were observed for mast cell subtypes. For subtyping of mast cells into tryptase+ and chymase-tryptase+ cells, defining granule morphology below the diffraction limit of light is crucial and can only be analyzed with EM[22]. Tryptase+ mast cell granule content is characterized by well-defined scrolls (Fig. 2c, d), whereas chymase-tryptase+ mast cells have more homogeneous granules (Fig. 2e, f). Over 90% of mast cells in the donors with type 1 diabetes were identified as tryptase+, while ~50% of total mast cells were tryptase+ for both autoantibody-positive and control groups

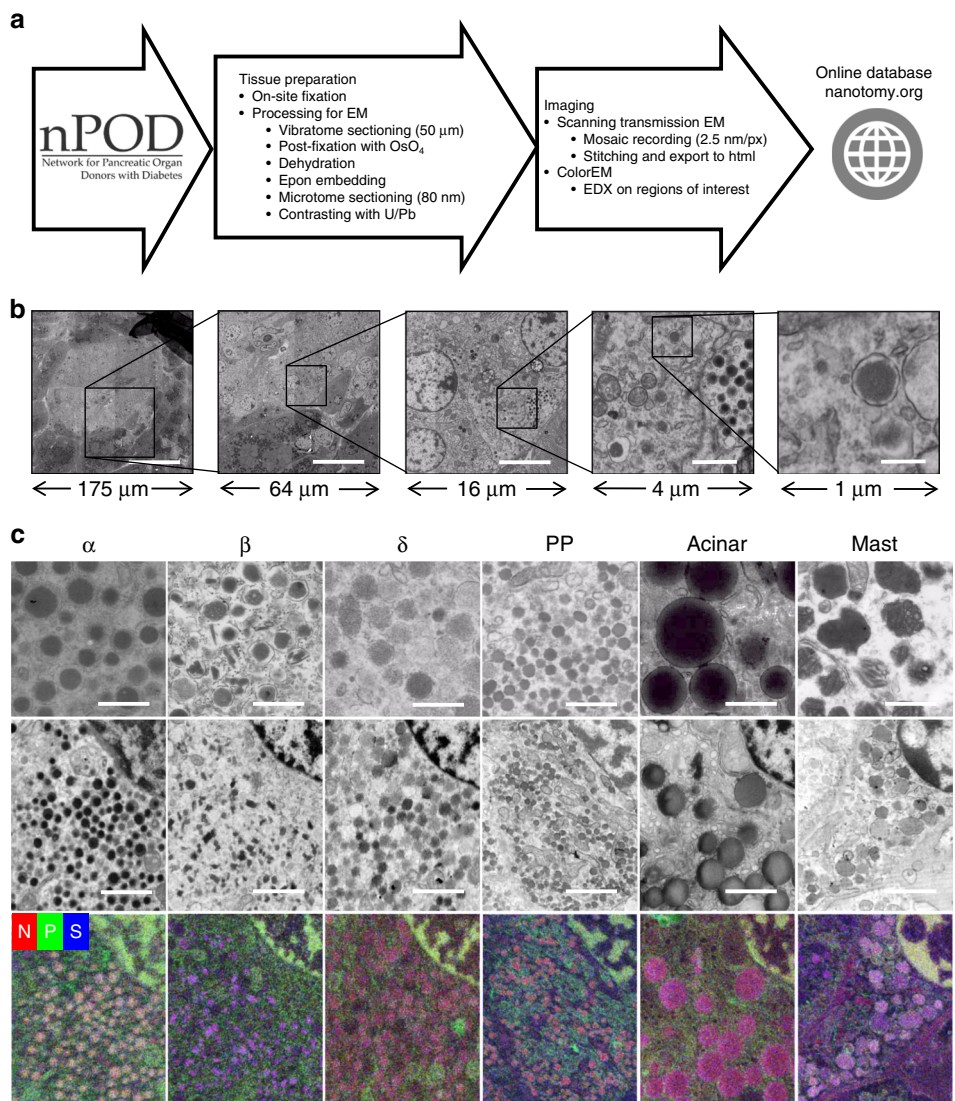

**Fig. 1 nPOD nanotomy followed by ColorEM allows zooming into islets up to macromolecular level and label-free identification. a** Fixed pancreatic samples are received from nPOD and processed for EM. Standard acquisition is at 2.5 nm pixel size. Stitched mosaics are converted to html and uploaded to www.nanotomy.org. Samples may be revisited for elemental analysis (ColorEM). **b** Overview and molecular detail with nanotomy can only be fully appreciated at www.nanotomy.org. For illustration purposes a step-wise example of islet overview (left), cells with subcellular details like the nucleus (middle); up to organelles including secretory vesicles (right) is shown. c Pancreatic cell types are discriminated based on granule morphology (top panels; see material and methods for description) and elemental composition (bottom row). EDX maps of phosphorus (green), nitrogen (red), and sulfur (blue) are overlaid. Granules of each cell type have high nitrogen (red). Granules of the α-cells are enriched in phosphorus (orange in overlay), β-cell granules with sulfur (purple in overlay), δ-, PP- and exocrine acinar cell granule show mainly nitrogen (red to pink in overlay), and mast cell granules contain sulfur (purple in overlay). Bars: (**b**, left to right) 50 μm, 20 μm, 4 μm, 1 μm, and 250 nm, **c** 0.5 μm (upper panels), 1 μm. (lower panels) Upper panel images from donors: 6331 (α), 6130 (β), 6126 (δ, acinar), 6130 (PP), 6087 (mast cell). ColorEM of donors 6126 (α, β, δ, and acinar) and 6130 (PP and mast). Micrographs in **b**, **c** are representative and similar results can be obtained from each dataset. Raw EDX data are shown in Supplementary Fig. 1.

(Fig. 2e–g). Mast cells are classically known for their role in allergies, but a broader role for mast cells in physiology and immunity is considered, including recruitment of neutrophils, and production of pro-inflammatory cytokines and chemokines[23]. A role for mast cells in type 1 diabetes pathogenesis was recently suggested as well[24], though the role they might play is still unknown. Moreover, ultrastructural mast cell subtyping was never performed before on type 1 diabetes pancreas samples, so the prominence of tryptase+ mast cells compared to control could suggest a disease-related role. Thus nPOD nanotomy analysis shows statistically significant differences in innate immune cell prevalence between type 1 diabetes and control donors.

**Intermediate cells observed in autoantibody-positive and type 1 diabetes donor tissue.** The division of endocrine and exocrine functions and topology of the pancreas is typically strict for secretion of hormones and digestive enzymes, respectively[13,14]. Furthermore, the ultrastructure of both pancreatic regions is distinct as determined from secretory granule morphology. However, unique 'intermediate cells' that contain both zymogen and hormone storage granules were identified in 2 of 16 (13%) control donors, 3 of 13 (23%) autoantibody-positive donors, and 6 of 16 (38%) type 1 diabetes donors (Fig. 3a–c). In most donors, the intermediate cells were located at the periphery of the islet (6301; Fig. 3c) while in some type 1 diabetes donors,

**Table 1 Mast cells and neutrophils enrichment in pancreas of nPOD donors.**

| Group | CaseID | Pancreas Region | AAb (RIA) | Age (years) | Duration (years) | Mast cells total | Mast cells T+ | Neutrophils |
|---|---|---|---|---|---|---|---|---|
| Non-diabetes | 6098a | Head | – | 18 | – | 4.5 | 0 | 0 |
| | 6098b | | | | | 0 | 0 | 0 |
| | 6126a | | | | | 3.2 | 0 | 0 |
| | 6126b | Head | – | 25 | – | 0 | 0 | 0 |
| | 6126c | | | | | 3.5 | 0 | 0 |
| | 6130 | Head | – | 5 | – | 4.8 | 4.8 | 0 |
| | 6131 | Head | – | 24 | – | 0 | 0 | 0 |
| | 6153 | Head | – | 15 | – | 0 | 0 | 0 |
| | 6160 | Head | – | 22 | – | 1.0 | 1.0 | 1.0 |
| | 6227 | Head | – | 17 | – | 0 | 0 | 0 |
| | 6229 | Head | – | 31 | – | 0 | 0 | 0 |
| | 6230 | Body | – | 16 | – | 0 | 0 | 0 |
| | 6232 | Head | – | 14 | – | 0 | 0 | 0 |
| | 6233 | Head | – | 14 | – | 0 | 0 | 0 |
| | 6331 | Body | – | 27 | – | 0 | 0 | 0 |
| | 6374 | Head | – | 14 | – | 0 | 0 | 0 |
| | 6412a | Head | – | 17 | – | 0 | 0 | 2.6 |
| | 6412b | | | | | 0 | 0 | 0 |
| | 6417a | Head | – | 18 | – | 0 | 0 | 0 |
| | 6417b | Body | | | | 11.2 | 0 | 0 |
| Pregnancy | 6226 | Head | – | 38 | – | 2.3 | 0 | 0 |
| | *Average* | | | | | 1.4 ± 0.6 | 0.8 ± 0.6 | 0.2 ± 0.1 |
| AAb+ | 6151 | Head | GADA | 30 | – | 2.1 | 0 | 0 |
| | 6156 | Head | GADA | 40 | – | 5.6 | 0 | 1.1 |
| | 6158a | Head | GADA, mIAA | 40 | – | 0 | 0 | 0 |
| | 6158b | | | | | 5.0 | 0 | 0.6 |
| | 6197 | Head | GADA, IA-2A | 22 | – | 0 | 0 | 0 |
| | 6301 | Body | GADA | 26 | – | 1.2 | 1.2 | 0 |
| | 6303 | Body | GADA | 22 | – | 1.7 | 1.7 | 0 |
| | 6314a | Head | GADA | 21 | – | 4.0 | 4.0 | 0 |
| | 6314b | Body | | | | 0.6 | 0.6 | 0 |
| | 6388 | Body | GADA, mIAA | 25 | – | 0 | 0 | 3.5 |
| | 6397 | Body | GADA | 21 | – | 2.4 | 2.4 | 0 |
| | 6400 | Body | GADA | 25 | – | 3.3 | 3.3 | 0 |
| | 6421 | Body | GADA | 6 | – | 0 | 0 | 0 |
| | 6424a | Head | GADA, mIAA | 17 | – | 7.8 | 7.8 | 0 |
| | 6424b | Body | | | | 5.0 | 5.0 | 9.9 |
| | 6433a | Body | GADA | 23 | – | 2.4 | 2.4 | 0 |
| | 6433b | | | | | 0 | 0 | 0 |
| | *Average* | | | | | 2.4 ± 0.6 | 1.7 ± 0.6 | 0.9 ± 0.6 |
| Type 1 Diabetes | 6063 | Head | – | 4 | 3 | 2.5 | 2.5 | 1.2 |
| | 6064 | Head | GADA, IA-2A, ZnT8A | 20 | 9 | 1.2 | 1.2 | 39.1* |
| | 6087a | Head | ZnT8A | 18 | 4 | 6.7 | 6.7 | 0 |
| | 6087b | | | | | 5.9 | 5.9 | 0 |
| | 6113 | Head | – | 13 | 1.6 | 0 | 0 | 12.8 |
| | 6198a | Tail | GADA, IA-2A, ZnT8A | 22 | 3 | 5.6 | 5.6 | 8.4 |
| | 6198b | | | | | 4.1 | 4.1 | 24.6 |
| | 6204a | Body | GADA | 28 | 21 | 0 | 0 | 21.7 |
| | 6204b | | | | | 0.8 | 0 | 21.7 |
| | 6209 | Head | IA-2A, ZnT8A | 5 | 0.25 | 0 | 0 | 0 |
| | 6211a | Body | GADA, IA2A, ZnT8A | 24 | 4 | 0.9 | 0.9 | 0 |
| | 6211b | | | | | 1.2 | 0 | 0.6 |
| | 6228a | Head | | | | 0 | 0 | 0 |
| | 6228b | | GADA, IA-2A, ZnT8A | 13 | 0 | 0 | 0 | 9.8 |
| | 6228c | Body | | | | 0 | 0 | 0 |
| | 6243a | Body | – | 13 | 5 | 3.4 | 3.4 | 1.7 |
| | 6243b | | | | | 2.4 | 2.4 | 2.4 |
| | 6337 | Head | – | 20 | 5 | 4.5 | 4.5 | 0 |
| | 6360 | Body | – | 5 | 2.5 | 0 | 0 | 0 |
| | 6362 | Body | GADA | 25 | 0 | 3.3 | 3.3 | 0 |
| | 6380 | Body | – | 11 | 0 | 1.3 | 1.3 | 0 |
| | 6405 | Head | GADA, IA2A, ZnT8 | 29 | 0.6 | 7.5 | 7.5 | 0 |
| | 6436a | Body | IA2A | 26 | 3 | 0 | 0 | 0 |
| | 6436b | Head | | | | 4.1 | 4.1 | 21.6* |
| | *Average* | | | | | 2.3 ± 0.5 | 2.2 ± 0.5 | 6.9 ± 2.2 |
| Type 2 Diabetes | 6124 | Head | – | 62 | 3 | 0 | 0 | 0 |
| | 6133 | Head | – | 46 | 20 | 1 | 0 | 3.8* |
| | *Average* | | | | | 0.6 ± 0.6 | 0 | 1.9 ± 1.3 |

Donors are sorted by condition and case-ID, and information on pancreas donor region, type of autoantibodies present as measured by radioimmunoassay (RIA), age of demise, and disease duration is provided. The number of total and tryptase+ (T+) mast cells and neutrophils per $10^5\,\mu m^2$ were recorded. The neutrophils observed were mostly in the exocrine region except those marked (*) had one or more in and/or around the islet. Innate immune cells average ± standard error are shown for each group.

the intermediate cells were found scattered throughout a remnant islet (for example, see donor 6063 in the database). EDX analysis showed high nitrogen content for both types of granules with an additional phosphorus signal in the endocrine granules in 6301 (autoantibody-positive) and 6228 (type 1 diabetes) donors (Fig. 3d lower panel and f), suggesting these contain glucagon, while intermediate cells in 6227 (control) and a subset in 6301 (autoantibody-positive) show sulfur-containing granules,

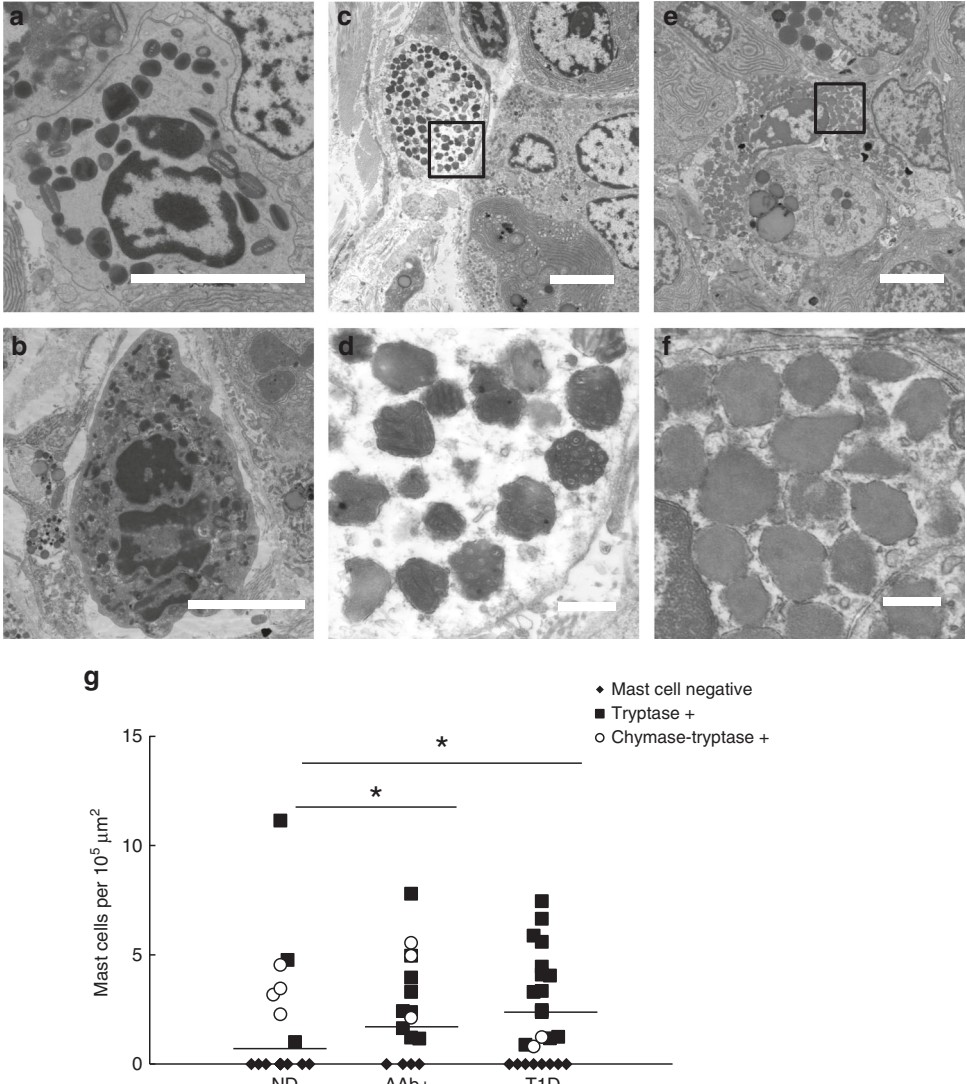

**Fig. 2 Eosinophils, neutrophils, and mast cells are prominent in type 1 diabetes donors.** Granulocytes from the nPOD data are distinguished based on secretory granule morphology. **a** Eosinophils were found in exocrine tissue of one donor. **b** Neutrophils are detected in multiple type 1 diabetes, predominantly located in the exocrine parenchyma and some in the peri-islet region, as shown here. Mast cells are observed in type 1 diabetes (**c**) and control samples (**e**). Tryptase+ mast cells are identified by prominent scrolls in the secretory granules (**d**), and chymase-tryptase+ mast cells show homogenous gray granules (**f**). Total numbers and prevalence of the innate immune cells per dataset are displayed in Table 1. **g** Mast cells present per $10^5$ $\mu m^2$ in the total dataset per donor, each symbol represents one donor. Average numbers of total mast cells did not significantly differ between control ($1.4 \pm 0.60$), autoantibody-positive ($2.4 \pm 0.57$; $p = 0.068$ versus control), and type 1 diabetes donors ($2.3 \pm 0.49$; $p = 0.062$ versus control). However, tryptase+ mast cells were found significantly higher in autoantibody-positive donors (AAb+; mean $= 1.7 \pm 0.55$, $p = 0.02$; $n = 17$) and type 1 diabetes donors (T1D; mean is $2.2 \pm 0.50$, $p = 0.005$; $n = 24$) than control non-diabetes donors (ND; $0.81 \pm 0.57$; $n = 21$). Means from tryptase+ mast cell numbers are indicated by horizontal lines. Statistical analyses between control, autoantibody positive, and type 1 diabetes donor groups were performed first by non-parametric one-way ANOVA resulting in tryptase+ mast cells ($p = 0.014$), followed by Mann–Whitney $U$ tests. (*) Significant differences. $n$ Indicates number of individual datasets analyzed per condition. Bars: 5 $\mu m$ (**a**–**c**, **e**) 0.5 $\mu m$ (**d**, **f**). Donors 6064 (**a**), 6380 (**b**), 6087 (**c**, **d**), and 6126 (**e**, **f**).

suggesting these contain insulin (Fig. 3b and d upper panel). Therefore, both morphology and EDX analysis indicated that intermediate cells contain endocrine as well as zymogen granules (Fig. 3, Supplementary Fig. 2).

The exocrine pancreas has received variable attention as a component potentially involved in type 1 diabetes pathogenesis (reviewed in ref. [20,21]). Type 1 diabetes patients show a significant reduction in pancreas weight or volume at the time of disease onset, and exocrine insufficiency has been reported[25–29]. Other findings include immunological alterations such as increased incidence of exocrine-specific autoantibodies[30,31], infiltration of immune cells in exocrine tissue[19,32], and complement activation localized to vessels and ducts throughout the exocrine tissue[33].

Furthermore, maintained β-cell mass but decreased pancreas volume among autoantibody-positive and even antibody-negative first-degree relatives[34,35], together with decreased pancreas weight from antibody-positive organ donors[27] indicate that exocrine tissue might be affected prior to clinically detectable changes in islet mass, composition, and function in type 1 diabetes. Intermediate cells, although with an unaffected morphology, have been observed in non-diabetic controls (Fig. 4a) and before in type 2 diabetes human islets, albeit with insulin granules[36,37]. Note the data using all the nPOD samples examined at this stage show a trend that these intermediate cells are more present in autoantibody+ and type 1 diabetes donors than with controls. Interestingly the intermediate cells observed in both the

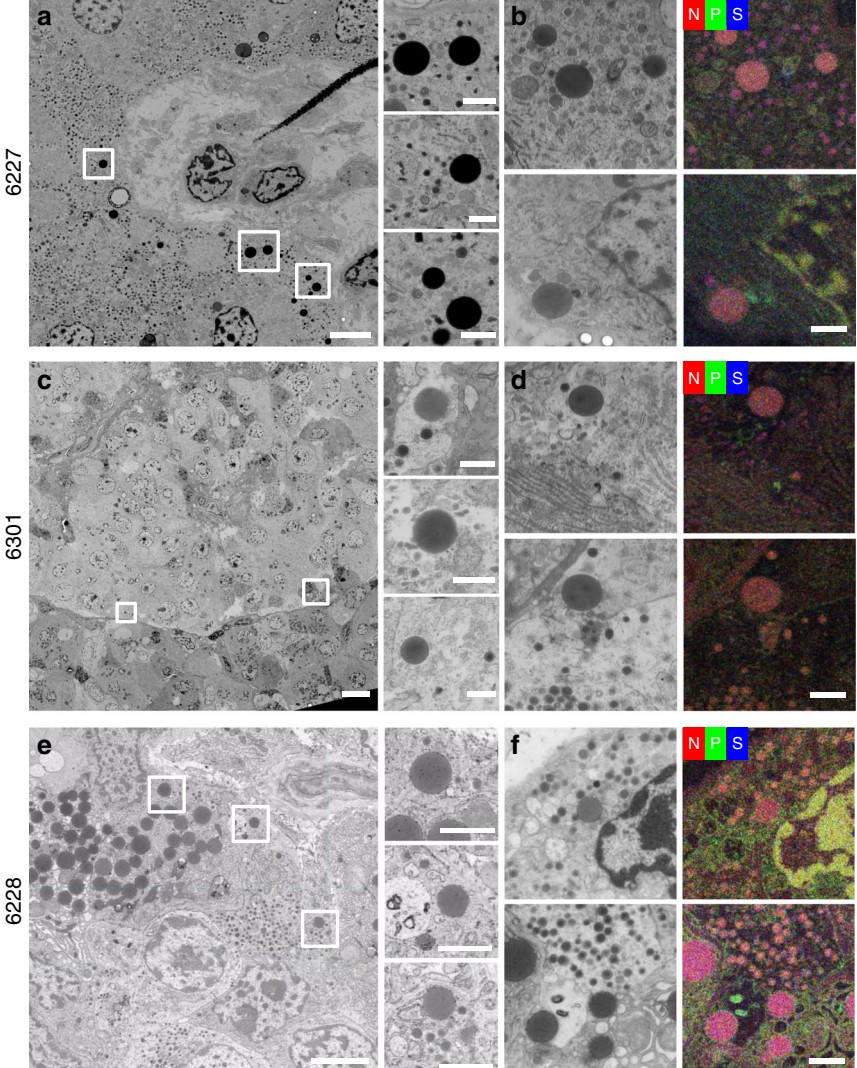

**Fig. 3 Abnormal endocrine-exocrine granules in the same cell relate to type 1 diabetes.** Cells containing both exocrine and endocrine granules were identified in the control (**a**, **b**; 6227; 2 of 16 donors), autoantibody-positive (**c**, **d**; 6301; 3 of 13 donors) and type 1 diabetes (**e**, **f**; 6228; 6 of 16 donors) donor groups, one example of each is shown here. The intermediate cells contain both secretory granules resembling exocrine and either insulin, in 6227 (**b**) and 6301 (**d** upper panel), or glucagon, in 6301 (**d** lower panel) and 6228 (**f**), granules based on morphology and elemental content using ColorEM with exocrine granules in red, insulin granules in purple, and glucagon granules in orange (see Fig. 1 for reference). Bars: 5 μm in overviews, 1 μm in boxed regions, and 1 μm in **b**, **d**, **f**. Raw EDX data are shown in Supplementary Fig. 2.

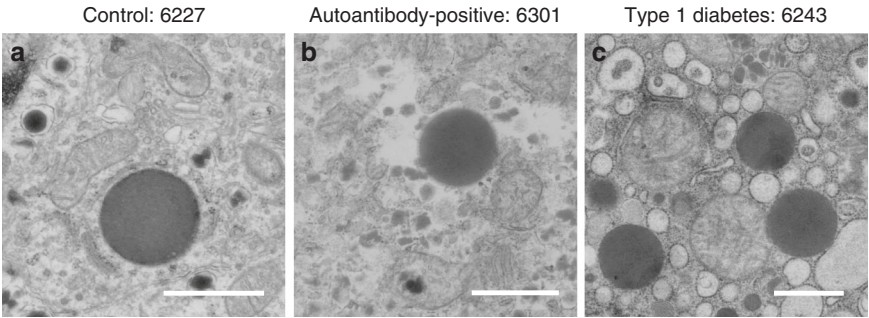

**Fig. 4 Only intermediate cells from autoantibody-positive and type 1 diabetes donors display an affected ultrastructure of mitochondria and ER.** **a** Intermediate cells from control donors do not display additional morphological alterations (n = 2), while autoantibody-positive (**b**; n = 3) and type 1 diabetes (**c**; n = 6) donors do show a cell stress associated affected ultrastructure looking at diminished mitochondrial cristae, dilation of rough ER, and extraction of material. n indicates the number of donors per group with intermediate cells. Scale bars: 1 μm.

autoantibody-positive and type 1 diabetes donors show affected morphology including dilated rough ER and diminished mitochondrial cristae (Fig. 4b, c). Similar abnormal appearing intermediate cells were previously found in a diabetes-prone rat model[11]. Therefore, additional dynamic methods and models are needed to determine whether this phenomenon is specifically related to type 1 diabetes pathogenesis or diabetes in general.

## Discussion

A large-scale EM repository containing tissue from human pancreas donors to study type 1 diabetes pathogenesis (nPOD nanotomy), is now available to investigate islets of Langerhans and surrounding exocrine regions up to macromolecular resolution while maintaining tissue context. Likewise, 3D high content EM methods are rapidly developing, like recent volumetric EM reconstructions of a *Drosophila melanogaster* brain and *C. elegans* nervous system[38,39]. A 3D dataset obtained through these efforts show valuable and detailed information on connectomics. Here, we focused on sampling many different donors focusing on large 2D datasets, making the nPOD large-scale EM repository suitable for disease-oriented research, in this case, for the study of type 1 diabetes. Open-access nanotomy data sharing, like initiatives on DNA sequences and protein structures, allows for the reuse of raw data in entirely different research questions. High resolution EM is typically used to generate qualitative data, due to the labor intensive image acquisition and limited sample size, but the nPOD nanotomy repository currently contains large-scale datasets, improving sample range up to sub-mm$^2$, from 47 donors and is still expanding. Moreover, nPOD nanotomy can be linked with quantitative approaches through the nPOD DataShare[40] focusing results and analysis of nPOD tissue of the same donors by other expert labs, including the use of proteomics[41], imaging mass cytometry[8,42], and many more techniques enabled by smart tissue biobanking (reviewed by ref. [4]). Next, automatic image analysis would be most valuable to further explore the repository, but this is still in development. While most progress is made on feature recognition in 3D datasets, where typically a subset of planes are manually annotated and the results are extrapolated in intermediate planes, such reference is absent in 2D image annotations. With the introduction of fast electron microscopes for section imaging[43–45] as well as dedicated 3D machines[46] the need for automated image analysis is a priority in the field, however still at the stage of development. For an overview of software approaches that may help automatic feature recognition of nanotomy datasets see ref. [47]. An alternative way for structure identification is the implementation of multimodal microscopy to identify features at the EM level with other imaging approaches[48], including EDX as shown in Fig. 3. Image analysis experts already found open-source nanotomy data to challenge their algorithms for automatic recognition. Therefore, the repository is not only valuable for diabetes research, but also aids image analysis approaches mentioned above and could set the stage for others to publish raw EM data.

Here, we focused on the analysis of mixed endocrine-exocrine cells, based on the co-appearance of their respective secretory granules within the same cells in type 1 diabetes and autoantibody-positive donor tissues. The subcellular appearance of both exocrine and endocrine granules in the same cell suggests a malfunction in either development of new cells[49], vesicle release, or cell degradation[50] and supports the increasing tendency suggesting that the exocrine pancreas might be involved in the type 1 diabetes pathogenesis[20,21]. While nPOD nanotomy is valuable to study human type 1 diabetes at the macromolecular level, kinetic follow-up studies of how exocrine and endocrine cells may be affected will require a model system that may be as basic as a zebrafish in which Islets of Langerhans can be dynamically analyzed and/ or cells can be manipulated in vivo[51].

The ultrastructure features of islets are only a mouse-click away for any investigator, bypassing laborious and expensive image-acquisition at other EM-core facilities. Thereby the unprecedented and still expanding EM nPOD nanotomy database marks a milestone in the sharing of raw, extensive, information dense data. nPOD nanotomy, in conjunction with complementary studies, will thereby expedite our expanding insight into the pathogenesis of human type 1 diabetes. The shared data will benefit the diabetes research community at large and will hopefully boost open-access image sharing from biobanked materials in other fields.

## Methods

**Donors**. Pancreas samples were recovered from donors with single ($N = 9$) or multiple ($N = 4$) autoantibodies and diabetes ($N = 16$ type 1 diabetes $N = 2$ type 2 diabetes) as well as from control donors ($N = 16$) matched for age, gender, and BMI as much as feasible (Supplementary Table 1). Sixty-four nanotomy islet datasets were created from these 47 donors. Donor information with demographic details and laboratory assays including HbA1c and C-peptide levels are listed in Supplementary Table 1. Detection of insulin-positive islets and other light microscopy histopathology findings are also listed in Supplementary Table 1. Samples were recovered following standard operating procedures including informed research consent by organ procurement organizations (OPOs) throughout the United States transplantation network and shipped to the nPOD Organ Processing and Pathology Core (OPPC) at the University of Florida for processing as previously described[4,52,53]. All experiments were conducted under the approval of the University of Florida Institutional Review Board and the current study fulfills all requirements as approved by the medical ethical review board of the University Medical Center Groningen.

**Pancreas sample electron microscopy processing**. Pancreas samples from the head, body, or tail regions (Supplementary Table 1) were minced into ~3 × 3 mm pieces and fixed in cold, freshly prepared 2% paraformaldehyde-1% glutaraldehyde for 48 h followed by transfer to phosphate-buffered saline for storage at 4 °C before batch shipment to the EM laboratory in the Netherlands[53]. Tissue vibratome sections (~50 μm; Microm HM 650V) were post-fixed in osmium tetroxide/ potassium ferrocyanide, followed by dehydration and flat-embedding in Epon as previously reported[9].

Semi-thin 1 μm sections were cut (UC7 ultramicrotome, Leica Microsystems, Vienna, Austria) and used to select regions with islets upon toluidine blue staining using light microscopy. Subsequent ultrathin (80 nm) sections were cut (UC7 ultramicrotome) and placed on formvar coated copper grids (Electron Microscopy Sciences, Hatfield, Pennsylvania). Finally, sections were contrasted with uranyl acetate followed by Reynold's lead citrate as previously described[9,54].

**EM acquisition, image processing, and nanotomy website**. Image data were acquired on a Supra 55 scanning EM (SEM; Zeiss, Oberkochen, Germany) using a scanning transmission EM (STEM) detector at 28 kV with 2.5 nm pixel size (2 nm pixel size for datasets 6098a and b, 6126a, 6197, 6087a, 6113, and 6198) with an external scan generator ATLAS 5 (Fibics, Ottawa, Canada) as previously described[10,54]. One dataset is made of multiple tiles with an average of 37 ± 4, one tile sizes 16k × 16k pixels. After image tile stitching, sample datasets were exported as an html file and uploaded to the website www.nanotomy.org/ (Fig. 1a). Nanotomy datasets in the website were organized by donor groups. Each image contains an islet with surrounding exocrine cells, or occasionally scattered endocrine cells through exocrine tissue in the case of type 1 diabetes, and thus permits ultrastructural analysis of multiple cell types, organelles, and macromolecules using a simple zoom process for any region or feature of interest (Fig. 1b).

**Cell types and image analysis**. Sample quality was checked using quality controls based membrane integrity and level of extraction of material resulting in blank spots in the images in the overall dataset. Each nanotomy dataset was manually screened, field by field, for the different cell types based on granule morphology for endocrine cells, acinar exocrine cells, mast cells, neutrophils, eosinophils, and aberrant cells (e.g., intermediate cells based on presence of both exocrine and endocrine granules). Cells in the pancreas could be discriminated by secretory granule morphology[13,14]. Glucagon-containing α-cell granules were the most electron-dense, typically 200–250 nm in diameter, and often contained a thin halo between the membrane and the electron dense core (Fig. 1c). Insulin-containing β-cell granules were less electron-dense and typically 250–300 nm in diameter and mature insulin granules had a more prominent halo between the membrane and a crystalline core (Fig. 1c). Granules of δ- and PP-cells were of variable electron-density and were differentiated mainly by size with granules 200–350 nm and 120–160 nm in diameter, respectively (Fig. 1c). In the exocrine pancreas, acinar

cells were characterized by abundant rough ER and larger zymogen-containing secretory granules (0.5–1.5 μm diameter; Fig. 1c). Eosinophils were observed by round-to-oval shaped granules with a well-defined dark or light core (Fig. 2a)[55]. Neutrophils were recognized by a variety of amorphous granules of different sizes and electron density, some of which had a dark core, and relatively dark cytoplasm (Fig. 2b)[56]. Mast cells were subdivided into tryptase+ and chymase-tryptase+ cells based on granule morphology. Tryptase+ mast cells were determined by amorphous secretory granules containing cylindrical clusters (Fig. 2c, d), while chymase-tryptase+ mast cells had homogeneous granules (Fig. 2f)[22]. Ultrastructural assessment was performed by analyzing mitochondrial cristae, dilation of rough ER as recognized by ribosomal lining, and extraction of material.

**Energy dispersive X-ray analysis (EDX; 'ColorEM')**. Energy dispersive X-ray analysis (EDX) imaging for elemental maps of phosphorus, nitrogen, and sulfur was performed as recently described[11]. Briefly, regions of interest were determined using the nanotomy maps. Next, serial sections of 100 nm were cut and placed on a formvar-coated single slot pyrolytic carbon grid (Ted Pella, INC., California, USA) followed by uranyl acetate staining. The selected regions of interest were imaged using an Oxford Instruments X-Max[N] 150 mm$^2$ Silicon Drift Detector and AZtecEnergy software (Abingdon, UK) mounted on the Zeiss Supra55 SEM. EDX images were generated (sum of 30–40 frames) with 50 μs dwell time at 15 kV acceleration voltage and 8.4 nA beam current. Image analysis and processing was done in Adobe Photoshop 19.1.5 and included a Gaussian blur of 1.5 pixel radius to the raw elemental maps followed by adjustments of white points for each color channel in a level adjustment layer, followed by +25 brightness and +50 contrast adjustment layer. Raw EDX data are shown in Supplementary Figs. 1 and 2. The AZtecEnergy project files are available through www.nanotomy.org.

**Statistics**. Data are presented as means ± SEM with $N$ = number of donors unless otherwise indicated. Statistical analyses between control, autoantibody positive, and type 1 diabetes donor groups were performed first by non-parametric one-way ANOVA resulting in significant differences for neutrophils ($p = 0.005$) and tryptase+ mast cells ($p = 0.014$), followed by Mann–Whitney $U$ tests using SPSS Statistics V25 (IBM, Armonk, New York, USA) to assess differences between individual groups.

**Reporting summary**. Further information on research design is available in the Nature Research Reporting Summary linked to this article.

## Data availability
All data is open access available via the repository website www.nanotomy.org. Data includes large-scale EM maps linked via donor numbers and raw EDX data can be downloaded via the link below the donor data list.

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

## Acknowledgements

We thank the donors and their families for donating tissues to support this research. We also thank the Organ Procurement Organizations that partner with nPOD to recover organ donor tissues as well as the JDRF nPOD staff members for providing pancreas EM samples. Organ Procurement Organizations (OPO) partnering with nPOD to provide research resources are listed at http://www.jdrfnpod.org//for-partners/npod-partners/. The content and views expressed are the responsibility of the authors and do not necessarily reflect an official view of nPOD. We thank Ruby Kalicharan (UMCG) for technical assistance and Jacob Hoogenboom (Delft, The Netherlands) for discussions on EDX. This research was performed with the support of the Network for Pancreatic Organ donors with Diabetes (nPOD; RRID: SCR_014641), a collaborative type 1 diabetes research project sponsored by JDRF (nPOD: 5-SRA-2018-557-Q-R to M.A.A.) and The Leona M. & Harry B. Helmsley Charitable Trust (2018PG-T1D053). This work was also supported by the JDRF (6-2006-1140 25-2013-268 to M.C.T.; 25-2012-770 to M.C.T. and B.N.G.G.); the National Institutes of Health (UC4 DK108132 to M.A.A.); The Netherlands organization for scientific research (ZonMW 91111.006; STW Microscopy Valley 12718 to B.N.G.G.) and the European Association for the Study of Diabetes (EASD; to B.N.G.G.).

## Author contributions

P.d.B. and N.M.P. performed the experiments, researched the data and wrote the paper, A.H.G.W. and J.K. performed the experiments, I.K. researched the data and contributed to discussion, M.A.A. and M.C.T. contributed to discussion and reviewed/edited the manuscript, B.N.G.G. conceived of the study and wrote the paper.

## Competing interests

The authors declare no competing interests.
