## [Peer Review File · Nature Communications]

Reviewers' comments:

Reviewer #1 (Remarks to the Author):

Large-scale electron microscopy database for human T1D -

Boer et al present the development of an electron microscopy database of pancreata from non-diabetic, auto-antibody positive donors and donors with T1D aimed to understand mechanisms of T1D. This is the first such electron microscopy repository created for pancreatic tissues and will be an important resource for future studies involving alterations in sub-cellular structures in pancreatic islet cells. The paper upon revision, has included only minor changes explaining why automated EM image analysis/quantification so far is still a work-in-progress and mast cell subtype differentiation that could only be done by EM. However, our major concerns still remain the same, that is,

i) The contribution of this paper in understanding the disease better is still very limited. So far it seems that the same findings could be obtained through conventional histological studies

Minor comments:

In Fig 2G, 3 data points presented for mast cell (tryptase+) are represented in squares for non-diabetic (values of each ~0.5, 5 and 12) providing a mean of 3+. However, the mean bar and the figure legend mention a mean of 0.81 for non-diabetic

Reviewer #2 (Remarks to the Author):

This manuscript describes a large-scale electron microscopy database for human type 1 diabetes. As a matter of principle, data sharing platforms such as this are very useful for the scientific community and a worthy effort. However, the manuscript describing this database does not make the case for the real utility of the current dataset. The authors' observations are incremental, somewhat over-interpreted and certainly not paradigm-shifting. Overall, the manuscript (and the database) does not make the case for the value that the authors ascribe to it, and there is nothing in the manuscript itself that makes it appealing to a broad readership.

Referee #1 (Remarks to the Author):

Comments on de Boer et al.

The present manuscript details the generation of an openly-accessible, shared database of EM images of pancreas sections from various autoimmune/non-AI pancreas donors, in an attempt to aid efforts to identify the mechanisms driving autoimmune destruction of pancreatic beta cells in T1D. This nano-anatomy (“nanotomy”) database allows analysis of pancreas sections at high or low resolution as desired, visualization of structures from large-sized to nanoscale, and limited elemental analysis (facilitated via energy dispersive X-ray analysis during image collection).

The paper adequately describes the technical approach taken to manufacturing the image database. However, beyond establishing a database of EM images as a community resource, the primary analysis carried out on the database does not truly provide any new insights into the etiology of T1D. Most, if not all of the features raised in the present manuscript have been previously described. The descriptive features mentioned in the paper, while potentially interesting, do not provide insights into whether they play any role in the disease process, or are merely caused by it; for example, they may represent tissue repair processes in which cells of the innate immune system are known to play a role. Without such new insights, it seems to me that this is a relatively minor technical advance that would be more interesting and important to circulate to a specialized audience of diabetes researchers rather than the general readership of Nature.

Indeed, the major goal is to present an EM repository, which is unique in its size of the single datasets and the amount of datasets. Here focused to aid the study of type 1 diabetes pathology as stressed in the title. We agree that the primary analyses are approached semi-quantitatively, but this is already a giant step for EM. Classical EM would only present snapshot images of user-selected representative regions and qualitative description of observations done behind the microscope. Our results illustrate how the step towards more quantification can be made, not only by the TEM operator but also by anyone with access to the open-source repository. This has become routine in other fields like DNA, proteomics, crystallographic data, but for EM this is the first open repository documenting large-scale data from a cohort >50 individuals, which can catalyze EM analysis like has been observed in these other fields.

Findings presented here do not provide insight in the dynamics of the disease progression, which is unfortunately not possible with any method since human donor material is always end-stage. We make this human material virtually accessible worldwide. Therefore, the strength of the repository is that novel findings can be linked to other methods to study the disease. We will include how this resource aids to other initiatives within nPOD in the discussion section.

Referee #2 (Remarks to the Author):

This manuscript reports on a the generation of a nanoscale image repository by electron microscopy of human donor islets. The work has been ongoing for a number of years and has used samples from the network for Pancreatic Organ Donors with Diabetes (nPOD).

Technologically, this is of course a tour-de-force of sample processing, image collection and data storage. The problem though is that it is not nearly as well demonstrated that the enormous amount of stored pixels can be processed into data that are representative of the available sample collection. The authors reference the need for ‘quantitative approaches through the nPOD DataShare’, and I think that the relevance of the present work will only be proven once high throughput, automated image analysis enables one to extract solid quantitative data.

We agree (and are exploring) that automated image analysis would be most valuable for the analysis of the repository, but for EM this is notoriously difficult. World-wide emphasis is on the development of automation, however the EM field is nearly not there yet. We will emphasize the need in the manuscript, and include the current state of the art in automated image analysis and how this may become useful to analyze the current datasets. As soon as such high-throughput analysis is possible the value of the database will only increase. For now, analysis can be performed in many ways for example by scoring and/or annotating events and structures of interest by anyone as long as you have a computer and internet connection, without working hands-on at a transmission EM recording single images per sample. Publishing our enormous amount of unique data from many individuals (i.e. not a single small snapshot of tissue from one or a few study subjects which is typically done) in an open source matter will actually expedite image analysis, including use of AI, by expert mathematicians and computer scientists.

And that brings us to the new insights that the current manuscript brings to support the usefulness of the repository, which I unfortunately find less convincing. I have the impression that the authors have gone in and looked at the images in a 'Google-earth' like way, zooming in on certain observations but likely missing out on many aspects of the bigger picture. As a consequence they end up with observations that could have often been made by ordinary histology techniques. The determination of neutrophil, eosinophil and mast cell numbers is an example. The intermediate cells have been documented before. What remains is the non-quantitative finding of somewhat dilated rough ER and diminished mitochondrial cristae.

We actually did use both overview *and* high resolution on samples that are not immuno-labeled. A few examples: (i) for the scoring of the different innate immune cells, we indicate where they localize within the tissue; (ii) in some donors we found intermediate exocrine and endocrine cells in remnant islets as described in the manuscript, which could not be observed if we did not keep the bigger picture.

We agree that for some of the immune cells described histology methods can also be performed. However, the manuscript aids in recognizing cell types, and the immune cells have a very distinctive morphology. While directly scoring, it provides insight in the prevalence and localization of these cells per donor. Moreover, for subtyping of mast cells, defining granule morphology at EM resolution is a crucial approach. Histological studies have shown mast cells before, but without subtype discriminating markers, which makes it a unique observation that tryptase+ mast cells are more prominently present in type 1 diabetes compared to control. Intermediate cells have been documented before, but never to this extent showing a more prominent link to autoantibody+ and type 1 diabetes donors than with controls. We will clarify this point in the results section.

In conclusion: this work needs its value established by an equally large scale image analysis effort prior to being considered for Nature, but it should be a very useful repository.

Publishing the work makes it available for researchers that may come in with the expertise and different questions mentioned by the reviewer on the current manuscript. Moreover, image analysis experts already found open source nanotome data to challenge their algorithms for automatic recognition. Therefore publishing the data as a repository will aid progress in diabetes research, image analysis as well as setting the stage for others to publish raw EM data (sampling sub-mm² size tissue) as opposed to sampling in the range of 100nm². The unprecedented availability of EM data that can be linked to other donor analysis methods ('EM analysis is just a push-button away and available worldwide' instead of laborious EM-preparation) had been stressed in the cover letter. We will emphasize this step forward in the revised manuscript.